# Synthesis and Evaluation of 3-Substituted-4-(quinoxalin-6-yl) Pyrazoles as TGF-β Type I Receptor Kinase Inhibitors

**DOI:** 10.3390/molecules23123369

**Published:** 2018-12-19

**Authors:** Li-Min Zhao, Zhen Guo, Yi-Jie Xue, Jun Zhe Min, Wen-Jing Zhu, Xiang-Yu Li, Hu-Ri Piao, Cheng Hua Jin

**Affiliations:** College of Pharmacy, Yanbian University, 977 Gongyuan Road, Yanji 133002, China; 13043384228@163.com (L.-M.Z.); guozhen0921521@163.com (Z.G.); 17111030018@fudan.edu.cn (Y.-J.X.); junzhemin23@163.com (J.Z.M.); wenjing077@163.com (W.-J.Z.); 15944282596@163.com (X.-Y.L.); piaohr@ybu.edu.cn (H.-R.P.)

**Keywords:** TGF-β, ALK5 inhibitor, kinase assay, selectivity, docking

## Abstract

The transforming growth factor-β (TGF-β), in which overexpression has been associated with various diseases, has become an attractive molecular target for the treatment of cancers. Thirty-two quinoxaline-derivatives of 3-substituted-4-(quinoxalin-6-yl) pyrazoles **14a**–**d**, **15a**–**d**, **16a**–**d**, **17a**–**d**, **18a**–**d**, **19a**–**d**, **25a**, **25b**, **25d**, **26a**, **26b**, **26d**, **27b**, and **27d** were synthesized and evaluated for their activin TGF-β type I receptor kinase and p38α mitogen activated protein (MAP) kinase inhibitory activity in enzymatic assays. Among these compounds, the most active compound **19b** inhibited TGF-β type I receptor kinase phosphorylation with an IC_50_ value of 0.28 µM, with 98% inhibition at 10 µM. Compound **19b** also had good selectivity index of >35 against p38α MAP kinase, with 9.0-fold more selective than clinical candidate, compound **3** (LY-2157299). A molecular docking study was performed to identify the mechanism of action of the synthesized compounds and their good binding interactions were observed. ADMET prediction of good active compounds showed that these ones possess good pharmacokinetics and drug-likeness behavior.

## 1. Introduction

Transforming growth factor-β (TGF-β) superfamily members have a wide range of cellular functions, including cell proliferation, differentiation, adhesion, migration, and apoptosis [1]. Moreover, TGF-β superfamily members are proteins with similar structures, including TGF-βs, activins, bone morphogeneticproteins (BMPs), growth and differentiation factors. TGF-β plays a crucial role in initiation and progression of fibrosis in various tissues such as the heart [2], lung [3], liver [4], and kidney [5]. TGF-βs are composed of five homogeneous isomers with highly homologous amino acid sequences, TGF-β1, TGF-β2, TGF-β3, TGF-β4, and TGF-β5, though only the first three exist in humans. Among these isoforms, TGF-β1 is the prototype and major isoform of this family. TGF-β conducts signaling through two distinct serine and threonine kinase receptors as TGF-β type I (activin receptor-like kinase 5, ALK5) and type II receptors [6]. ALK5 is activated by the combination of TGF-β and the type II receptor in the juxtamembrane GS domain, stimulating its kinase activity. The activated ALK5 spread the signals through phosphorylation of Smad2 and Smad3, and followed by binding with Smad4 to form complexes. These Smad complexes will translocate into the nuclei, where they regulate target gene transcription such as cell differentiation, proliferation, apoptosis, migration, and extracellular matrix production [1]. Nevertheless, overexpression of TGF-β signaling was shown to result in various human diseases such as hematological malignancy [7], cancer [8], and pancreatic diseases [9].

For this reason, many small molecule ALK5 inhibitors, such as compounds **1** (SB-505124) [10], **2** (SD-208) [11], **3** (LY-2157299) [12], and **4** (EW-7197) [13] were synthesized at major research institutions (Figure 1). These compounds inhibited ALK5 autophosphorylation and TGF-β-induced transcription of extracellular matrix genes at sub-micromolar concentrations in reporter assays, as shown in Figure 1. Among them, clinical candidates, compounds **3** and **4** have progressed to Phase II and Phase I trials as antitumor agents, respectively.

We previously showed that a series of compounds, denoted as **5**, containing the quinoxaline moiety, except for the 2,3-dimethyl substituted analogs, showed significant ALK5 inhibition in enzymatic assay [14]. P38α MAP kinase domain is among the most homologous to that of ALK5, so we selected p38α MAP kinase to evaluate the selectivity profile of these series of compounds. In addition, previous studies have found that compounds with high p38α MAP kinase selectivity are also highly selective to other kinases, such as compound **4**, which showed high selectivity to 320 kinases in human body [13]. These series of compounds were selective for ALK5, compared with p38α MAP kinase. The most active compound inhibited ALK5 phosphorylation with an IC_50_ value of 0.013 µM and a selectivity index of >77 against p38α MAP kinase. We reported the effect of quinoxaline groups and length of side chains attached to pyrazole ring, and positional isomers on the activity of compounds containing pyrazole rings, but the effect of the pyridine portion attached to pyrazole ring on the activity of the compounds was not discussed.

Tojo et al. described a novel class of ALK5 inhibitors possessing a thioamide linkage between the phenyl and pyrazole rings [15]. Among these, compound **6** (A-83-01) inhibited ALK5 with an IC_50_ value of 0.012 µM. Although including a thioamide linkage between the phenyl and pyrazole ring distinctly increased ALK5 inhibitory activity, as previously shown [16], the thioamide linkage was rather unstable and was slowly cleaved, to release a pyrazole ring, during long-term storage.

Previously, we showed that the methyl group of 6-methylpyridine in compound **4** formed hydrophobic interactions with the aromatic ring of Tyr249 and the nitrogen atom of the same moiety formed a water-mediated hydrogen bonding network with the side chains of Tyr249 and Glu245 and the backbone of Asp351 [13]. In previous studies, we found that the introduction of electron-donating groups into quinoxaline or quinoline rings failed to increase ALK5 inhibition activity [14]. This is due to the steric hindrance of their rigid structures. Therefore, we assumed that the ALK5 inhibition activity will be enhanced by introducing the electron-donating groups or the structure with many binding sites into the non-rigid pyridine ring on the premise of maintaining the non-substituent quinoxaline structure. To examine whether the capability of the nitrogen atom of the 6-methylpyridine moiety as an H-bond acceptor would be increased by other substitutions, we introduced 6-(dimethylamino)pyridin-2-yl, 4-methylthiazol-2-yl [17,18,19], and pyrimidin-4-yl groups [20], instead of the 6-methylpyridine moiety in **5** series compound. Based on this finding and previous research, we tried to replace the thioamide linkage with a chemically stable thioamidomethylene linkage and, designed compounds **18a**–**d**, **19a**–**d**, **27b**, and **27d** (Figure 2). To compare the effects of the thioamidomethylene linkage in **18a**–**d**, **19a**–**d**, **27b**, and **27d** on ALK5 inhibitory activity, their counterpart derivatives **14a**–**d**, **15a**–**d**, **16a**–**d**, **17a**–**d**, **25a**, **25b**, **25d**, **26a**, **26b**, and **26d** possessing an amidomethylene linkage, were also designed. The target compounds **14b**–**d**, **16b**–**d**, **15b**–**d**, **17b**–**d**, **18b**–**d**, **19b**–**d**, **25b**, **25b**, **25d**, **26b**, **26b**, **26d**, **27b**, and **27d** each possess a substituent, either *o*-F, *m*-F or *m*-CN, in the phenyl ring because these were previously found to be most beneficial for ALK5 inhibitory activity and selectivity [13].

## 2. Results and Discussion

### 2.1. Chemistry

The 3-(6-(dimethylamino)pyridin-2-yl)-4-(quinoxalin-6-yl)pyrazoles **14a**–**d** and 3-(4-methylthiazol-2-yl)-4-(quinoxalin-6-yl)pyrazoles **16a**–**d** were synthesized as shown in Scheme 1. The 6-(dimethylamino)picolinaldehyde (**8**) [21] and 4-methylthiazole-2-carbaldehyde (**9**) were treated with aniline and diphenyl phosphite in *i*-PrOH at room temperature to give the (phenylamino)methylphosphonates **10a** and **10b** in 90% and 70% yields, respectively. Coupling of the **10a** and **10b** with quinoxaline-6-carbaldehyde [22] in a mixture of THF and *i*-PrOH (4:1) at room temperature in the presence of Cs_2_CO_3_, followed by hydrolysis with 1 N HCl, produced the corresponding monoketones **11a** and **11b** in 71% and 52% yields, respectively [14]. Treatment of **11a** and **11b** with *N,N*-dimethylformamide dimethyl acetal (DMF•DMA) in *N,N*-dimethylformamide (DMF) at 80 °C, followed by cyclization with hydrazine monohydrate in absolute EtOH, produced the pyrazoles **12a** and **12b** in 68% and 71% yields, respectively [23]. The pyrazoles **12a** and **12b** were alkylated with 2-chloro-*N*-phenylacetamide (**13a**) [24], 2-chloro-*N*-(2-fluorophenyl)acetamide (**13b**), 2-chloro-*N*-(3-fluorophenyl)acetamide (**13c**) or 2-chloro-*N*-(3-cyanophenyl)acetamide (**13d**) [25] in the presence of NaH in anhydrous DMF to yield the target compounds **14a**–**d**, **16a**–**d** and their positional isomers **15a**–**d** and **17a**–**d** in 40–81% and 7–15% yields, respectively. The positional isomers were separated by column chromatography and their structures were confirmed by nuclear Overhauser enhancement (NOE) experiments. In NOE experiments, irradiation of the methylene protons of compound **14a** at *δ* 5.06 gave an enhancement of the proton *H*-5 in the pyrazole ring at *δ* 7.80, while irradiation of the methylene protons of compound **15a** at *δ* 5.17 gave no enhancement of the proton *H*-5 in the pyrazole ring at *δ* 8.02, confirming the respective alkylation positions.

Thionation of compounds **14a**–**d** and **16a**–**d** with Lawesson’s reagent in anhydrous 1,2-dimethoxyethane (DME) at 85 °C produced the thioamides **18a**–**d** and **19a**–**d** in 37–89% yields as shown in Scheme 2.

To increase binding sites with key proteins, the 3-(pyrimidin-4-yl)-4-(quinoxalin-6-yl)pyrazoles **25a**, **25b**, and **25d** were synthesized as shown in Scheme 3. Pyrimidine-4-carbaldehyde (**20**) [26] was synthesized from commercially available 1,1-dimethoxyacetone and *N*,*N*-dimethylformamide dimethyl acetal via 3 steps. Compound **23** was synthesized from compound **20** via 3 steps in the same reaction condition as described in Scheme 1. The pyrazole **23** was further alkylated with substituted phenylacetamides **24a**, **24b**, or **24d** in the presence of NaH in anhydrous DMF to yield the target compounds **25a**, **25b**, **25d** and their positional isomers **26a**, **26b**, **26d** in 58–65% and 7–13% yields, respectively. These positional isomers were also separated by column chromatography and their structures were confirmed by NOE experiments (Scheme 3). Similarly, the thioamide compounds **27b** and **27d** were synthesized from **25b** and **25d** in the same reaction condition as described in Scheme 2 (Scheme 4). As expected, all synthesized target compounds were quite stable during long-term storage at room temperature.

### 2.2. Residual Activity in an Enzymatic Assay

To investigate whether compounds **14a**–**d**, **15a**–**d**, **16a**–**d**, **17a**–**d**, **18a**–**d**, and **19a**–**d** would inhibit ALK5, a kinase assay for preliminary screening was performed using the purified human ALK5 kinase domain produced in Sf9 insect cells and compounds at 10 μM. To examine the activity and potency of the synthesized compounds, compound **3** (LY-2157299) was used as a positive control. All compounds with a 4-methylthiazol-2-yl moiety (**16a**–**d**, **17a**–**d**, and **19a**–**d**) showed potent ALK5 inhibitory activity (27–98%), whereas those with a 6-(dimethylamino)pyiridin-2-yl moiety (**14a**–**d**, **15a**–**d**, and **18a**–**d**) showed moderate ALK5 inhibition activity (5–71%) (Table 1).

The amides **14a**–**d** (5–63%) and **16a**–**d** (95–97%) showed more potent ALK5 inhibitory activity than their respective positional isomers, **15a**–**d** (5–13%) and **17a**–**d** (27–54%), respectively. Among compounds containing a 6-(dimethylamino)pyridin-2yl moiety, the thioamides **18a**–**d** (30–71%) showed more potent ALK5 inhibition than the corresponding amides **14a**–**d** at 10 μM. Among compounds containing a 4-methylthiazol-2-yl moiety, the thioamides **19a**–**d** (87–98%) also showed similar ALK5 inhibition with the corresponding amides **16a**–**d** at 10 μM. We speculated that insertion of electron-donating groups at the 6-position of the pyridine moiety in **5** series compound would increase the capacity of the nitrogen atom in that moiety as an H-bond acceptor, thus, potentiating its ALK5 inhibitory activity. Instead, insertion of the 6-(dimethylamino)pyridin-2-yl moiety does not seem to fit ATP binding pocket of ALK5 compared to its structural counterparts bearing 6-methylpyrine. Fortunately, introduction of 4-methylthiazol-2-yl moiety effectively improved ALK5 inhibitory activity.

### 2.3. p38a MAP Kinase Assay

We selected p38α MAP kinase to survey the selectivity profile of this series of compounds because its kinase domain is among the most homologous to that of ALK5 [27]. All target compounds except **17a**–**d** (3–46%) did not inhibit p38α MAP kinase, even at their maximum concentration of 10 μM (Table 1).

Figure 3 intuitively illustrates the inhibitory activity of 3-substituted-4-(quinoxalin-6-yl)pyrazoles against ALK5 and p38α MAP kinase. All compounds with a 4-methylthiazol-2yl moiety (**16a**–**d**, **17a**–**d**, and **19a**–**d**) showed more potent ALK5 inhibition than those with a 6-(dimethylamino)pyridin-2-yl moiety (**14a**–**d**, **15a**–**d**, and **18a**–**d**).

### 2.4. ALK5 Inhibitory Activity in an Enzymatic Assay

In previous studies, we found that the activity of thioamide compounds was superior to that of the corresponding amide ones [14]. To evaluate ALK5 inhibitory activity and selectivity of the compounds possessing 6-(dimethylamino)pyridin-2-yl or 4-methylthiazol-2-yl moieties as electron donating group, the thioamides **18a**–**d** and **19a**–**d** was selected and their half maximal inhibitory concentration (IC_50_) values were measured. All compounds with a 4-methylthiazol-2-yl moiety (**19a**–**d**) showed potent ALK5 inhibition (IC_50_ = 0.28–0.57 μM), whereas those with a 6-(dimethylamino)pyridin-2-yl moiety (**18a**–**d**) showed no significant ALK5 inhibitory activity at up to 5.0 μM (Table 2).

To evaluate ALK5 inhibitory activity and selectivity of the compounds possessing pyrimidin-4-yl moiety as multiple binding site, the amides **25a**, **25b**, **25d** and thioamides **27b** and **27d** were also selected and evaluated. However, all compounds with a pyrimidin-4-yl moiety (**25a**, **25b**, **25d**, **27b**, **27d**) also showed no significant ALK5 inhibition activity at up to 2.26 μM (Table 2).

Compound **19b** showed the most potent ALK5 inhibitory activity with an IC_50_ value of 0.28 μM in these three series of compounds. It was slightly less potent than compounds **3** (0.12 μM). Furthermore, all thioamides **18a**–**d**, **19a**–**d**, **27b**, and **27d** failed to inhibit p38α MAP kinase up to 10.0 μM. Compound **19b** was the most selective in these three series, showing a selectivity index of >35, higher than that of positive control compound **3** (4). In this series of compounds (**19a**–**d**), the activity of compounds with substituents is superior to that of unsubstituted one. Notably, 2-fluorine substituted compound **19b**, which is two-fold more potent than unsubstituted compound **19a** (IC_50_ = 0.57 μM).

### 2.5. Docking Study of 18b and 19b in the Alk5 Active Site

To rationalize the SAR shown in Table 1 and Table 2, we examined the binding modes of two representative ligands (**18b** and **19b**) using the semi-flexible molecular docking program DS CDOCKER [28]. Docking analyses were performed using the recently reported X-ray structure of ALK5 complexed to a pyrazole ALK5 inhibitor (PDB: 1RW8) [13] (Figure 4).

The sulfur atom of the thioamide in **19b** contacted the hinge of ALK5, forming hydrogen bonds with the imidazole ring of His283, a residue previously reported to be important for inhibitory activity (C) [14]. The phenyl ring of **19b** interacted with Lys232 via Pi-alkyl bond. The central pyrazole ring of **19b** formed Pi-alkyl bond with the side chains of Leu340 and Val219. The thiazole N atom of **19b** formed carbon–hydrogen bond with the backbone of Ser287 and the methyl group of **19b** formed alkyl bond with the backbone of Lys337 and Pi-alkyl bond with the backbone of Phe289. Not only the calculated binding energy scores (CDOCKER_INTERATION_ENERGY) of these two compounds indicated that **19b** (−56.18 kcal/mol) formed more stable complexes with ALK5 than did **18b** (−54.81 kcal/mol), but also compound **19b** (Lys232, His283, Ser287, Leu340, and Lys337) showed more bonding with previously reported key amino acids than did compound **18b** (Lys232 and Leu340) (A) [29]. In particular, compound **18b** did not form bond with the most important amino acid, His283. The 2-fluorophenyl ring of **18b** and **19b** was stretched to the backside hydrophobic pocket consisting of Lys232, Leu260, Leu278, Val231, and Ala230. Furthermore, compound **19b** seemed to be more favorably accommodated in the binding pocket of ALK5 than compound **18b** (B and D). Our docking results indicated that the most active compound, **19b**, showed the more favorable intermolecular interactions in the ALK5 active site than compound **18b**. This supported the conclusion that the substitution group size of the pyridine moiety and selection of a heterocycle in compound **5** may have been important for improving ALK5 inhibition.

### 2.6. ADMET Analysis

ADMET pharmacokinetics is a very important method in drug design and drug screening, which is responsible for drug failure [30,31]. The ADMET properties of the drug molecules are greatly influenced by the optimum value of the intestinal absorption, water solubility, blood–brain barrier (BBB) penetration, human cytochrome P450 2D6 (CYP2D6) inhibition, and plasma protein binding (PPB) level. The ADMET parameters of these good targeted compounds **19a**–**d** was measured using Discovery Studio software as a drug reference was reported in Table 3.

The preferred and most widely used route of drug is the oral route, and the mechanism of absorption from the gastrointestinal tract is passive diffusion through the intestinal epithelial cells. Hence, the absorption and solubility of the drug are two major factors for oral administration. All of the 3-substituted-4-(quinoxalin-6-yl) pyrazoles **19a**–**d** showed good intestinal absorption. All compounds showed low or very low aqueous solubility at room temperature. However, the structure of these compounds contain thiazole and quinoxaline moiety, so it is easy to make a salt in stomach acid and dissolve in water. The BBB is an important organizational structure to maintain the stability of the central nervous system, which maintains the relative stability of the environment in the nervous system by restricting the entry of compounds into the central nervous system, and protects nerve cells from being invaded by harmful substances. All compounds, except compound **19b**, showed BBB penetration in permissible level (1). These compounds are suitable for the treatment of systemic diseases. Compound **19b** showed low BBB permeability and is suitable for non-brain diseases, which is characterized by cyano group at 3-position on the phenyl ring. In addition, the PPB binding ability of all compounds was good. CYP2D6 is an important drug metabolism enzyme in the family of cytochrome P450, and its catalysis is widely used. Over the years, the genes encoding CYP2D6 enzyme have been closely related to the genetic polymorphism, drug metabolism, production of adverse drug reactions, and activation of carcinogens. Also, all compounds did not inhibit CYP2D6, so they will be shown no or low side effects such as drug-drug interaction and wide metabolism. All of the parameters were within the acceptable range defined for human use and these good targeted compounds may exhibit significant pharmacokinetic and drug-likeliness properties.

## 3. Experimental

### 3.1. Chemistry

All solvents and chemicals (Aladdin, Shanghai, China) were commercially available without further purification. In general, all reactions were performed under normal atmosphere and at room temperature unless otherwise noted. Melting points were measured in open glass capillaries tube in an electrical melting point and are uncorrected. Spots were detected by viewing under UV lamps (254 nm). ^1^H and ^13^C NMR spectra were recorded on Bruker NMR spectrometers (Bruker, Billerica, MA, USA) at 300 MHz and 500 MHz, respectively, tetramethylsilane (TMS) was used as internal standard. High resolution mass spectra electrospray ionization (HRMS-ESI) was obtained on a Thermo Scientific LTQ Orbitrap XL spectrometer (Thermo Scientific, Markham, Ontario, Canada). The purity of the tested compounds was determined using an Agilent 1260 series HPLC system (Agilent Technologies, Waldbronn, Germany) using a C_18_ column (packing ODS HG 5 µM, 4.6 × 250 mm), and that for all the compounds was found to be >95%.

#### 3.1.1. General procedure for the preparation of diphenyl ((6-(dimethylamino)pyridin-2-yl)(phenylamino)methyl)phosphonate (**10a**), Diphenyl ((4-methylthiazol-2-yl)(phenylamino)methyl)phosphonate (**10b**) and Diphenyl ((phenylamino)(pyrimidin-4-yl)methyl)phosphonate (**21**)

To a stirred solution of **8**, **9**, or **20** (12.90 mmol) in *i*-PrOH (40 mL), aniline (15.48 mmol) and diphenyl phosphite (20.64 mmol) were added. The mixture was stirred at room temperature for 4 h. The reaction mixture evaporated to dryness under reduced pressure. The residue was purified by silica gel column chromatography (Petroleum ether/Ethyl acetate, 6:1) to give the titled compound **10a, 10b**, or **21** as a white solid.

*Diphenyl ((6-(dimethylamino)pyridin-2-yl)(phenylamino)methyl)phosphonate* (**10a**): Yield 90%; ^1^H NMR (300 MHz, CDCl_3_) *δ* 7.42 (t, *J* = 9.0 Hz, 1H), 7.36–7.21 (m, 6H), 7.18–7.08 (m, 6H), 6.86–6.77 (m, 4H), 6.42 (dd, *J* = 9.0, 3.0 Hz, 1H), 5.56 (br s, 1H), 5.27 (d, *J* = 18.0 Hz, 1H), 3.09 (s, 6H).

*Diphenyl ((4-methylthiazol-2-yl)(phenylamino)methyl)phosphonate* (**10b**): Yield 70%; ^1^H NMR (300 MHz, CDCl_3_) *δ* 7.36–7.14 (m, 10H), 7.04 (d, *J* = 9.0 Hz, 2H), 6.85–6.80 (m, 2H), 6.73 (d, *J* = 9.0 Hz, 1H), 6.21 (br s, 1H), 5.58 (d, *J* = 24.0 Hz, 1H), 2.41 (s, 3H).

*Diphenyl ((phenylamino)(pyrimidin-4-yl)methyl)phosphonate* (**21**): Yield 31%; ^1^H NMR (300 MHz, CDCl_3_) *δ* 9.23 (s, 1H), 8.68 (d, *J* = 3.0 Hz, 1H), 7.62 (s, 1H), 7.27–7.01 (m, 13H), 6.79 (t, *J* = 9.0 Hz, 1H), 6.70 (d, *J* = 9.0 Hz, 1H), 6.09 (br s, 1H), 5.30 (d, *J* = 24.0 Hz, 1H).

#### 3.1.2. General Procedure for the Preparation of 1-(6-(Dimethylamino)pyridin-2-yl)-2-(quinoxalin-6-yl)ethanone (**11a**) 1-(4-Methylthiazol-2-yl)-2-(quinoxalin-6-yl)ethanone (**11b**) and 1-(Pyrimidin-4-yl)-2-(quinoxalin-6-yl)ethan-1-one (**22**)

To a stirred solution of **10a**, **10b**, or **21** (10.9 mmol) in a mixture of THF (23.2 mL) and *i*-PrOH (5.8 mL), Cs_2_CO_3_ (1.41 mmol) and quinoxaline-6-carbaldehyde (10.9 mmol) were added. The mixture was stirred at room temperature for 16 h, and to it, 1 N HCl (43.4 mL) was added dropwise over a period of 5 min. The reaction mixture was diluted with *tert*-butyl methyl ether (MTBE) (17.4 mL). The aqueous layer was separated, and the organic layer was extracted with 1 N HCl (3 × 50 mL). The combined aqueous layer was neutralized with saturated NaHCO_3_ solution (pH 7–8) and extracted with EtOAc (3 × 100 mL). The EtOAc solution was dried over anhydrous Na_2_SO_4_, filtered, and evaporated to dryness under reduced pressure. The residue was purified by silica gel column chromatography (Petroleum ether/Ethyl acetate, 4:1) to give the titled compound **11a**, **11b**, or **22** as a yellow solid.

*1-(6-(Dimethylamino)pyridin-2-yl)-2-(quinoxalin-6-yl)ethanone* (**11a**): Yield 71%; ^1^H NMR (300 MHz, DMSO-*d*_6_) *δ* 8.92 (d, *J* = 3.0 Hz, 2H), 8.06 (d, *J* = 9.0 Hz, 1H), 8.01 (s, 1H), 7.80 (d, *J* = 9.0 Hz, 1H), 7.69 (t, *J* = 7.5 Hz, 1H), 7.22 (d, *J* = 9.0 Hz, 1H), 6.94 (d, *J* = 6.0 Hz, 1H), 4.76 (s, 2H), 3.15 (s, 6H).

*1-(4-Methylthiazol-2-yl)-2-(quinoxalin-6-yl)ethanone* (**11b**): Yield 52%; ^1^H NMR (300 MHz, DMSO-*d*_6_) *δ* 8.95 (br s, 2H), 8.08 (d, *J* = 9.0 Hz, 2H), 7.88 (s, 1H), 7.82 (d, *J* = 9.0 Hz, 1H), 4.81 (s, 2H), 2.54 (s, 3H).

*1-(Pyrimidin-4-yl)-2-(quinoxalin-6-yl)ethan-1-one* (**22**): Yield 62%; ^1^H NMR (300 MHz, CDCl_3_) *δ* 9.45 (s, 1H), 9.01 (d, *J* = 6.0 Hz, 1H), 8.83 (s, 2H), 8.11–8.06 (m, 2H), 7.94 (d, *J* = 3.0 Hz, 1H), 7.75 (d, *J* =9.0 Hz, 1H), 4.79 (s, 2H).

#### 3.1.3. General Procedure for the Preparation of *N*,*N*-Dimethyl-6-(4-(quinoxalin-6-yl)-1*H*-pyrazol-3-yl)pyridin-2-amine (**12a**) 4-Methyl-2-(4-(quinoxalin-6-yl)-1*H*-pyrazol-3-yl)thiazole (**12b**) and 6-(3-(Pyrimidin-4-yl)-1*H*-pyrazol-4-yl)quinoxaline (**23**)

To a stirred solution of **11a**, **11b**, or **22** (1.71 mmol) in anhydrous DMF (4.5 mL), *N*,*N*-dimethylformamide dimethyl acetal (5.12 mmol) were added. The mixture was heated at 80 °C for 4 h. After cooled to room temperature, the reaction mixture was evaporated to dryness under reduced pressure. The residue was dissolved in EtOH (6.43 mL), and to it, hydrazine monohydrate (35.36 mmol) was added. The mixture was heated at reflux temperature for 4 h, then cooled to room temperature, and evaporated to dryness under reduced pressure. The residue was diluted with CH_2_Cl_2_ (60 mL) and washed with water (20 mL) and brine (20 mL). The CH_2_Cl_2_ solution was dried over anhydrous Na_2_SO_4_, filtered, and evaporated to dryness under reduced pressure. The residue was purified by silica gel column chromatography (Petroleum ether/Ethyl acetate, 1:1) to give the titled compound **12a**, **12b**, or **23** as a yellow solid.

*N,N-Dimethyl-6-(4-(quinoxalin-6-yl)-1H-pyrazol-3-yl)pyridin-2-amine* (**12a**): Yield 68%; ^1^H NMR (300 MHz, CDCl_3_) *δ* 8.84 (d, *J* = 6.0 Hz, 2H), 8.21 (d, *J* = 3.0 Hz, 1H), 8.10 (d, *J* = 9.0 Hz, 1H), 7.88 (dd, *J* = 9.0, 3.0 Hz, 1H), 7.77 (s, 1H), 7.32 (t, *J* = 9.0 Hz, 1H), 6.68 (d, *J* = 9.0 Hz, 1H), 6.48 (d, *J* = 9.0 Hz, 1H), 3.10 (s, 6H).

4*-Methyl-2-(4-(quinoxalin-6-yl)-1H-pyrazol-3-yl)thiazole* (**12b**): Yield 71%; ^1^H NMR (300 MHz, CDCl_3_) *δ* 8.87 (d, *J* = 3.0 Hz, 2H), 8.29 (d, *J* = 3.0 Hz, 1H), 8.15 (d, *J* = 9.0 Hz, 1H), 7.95 (dd, *J* = 9.0, 3.0 Hz, 1H), 7.87 (s, 1H), 6.87 (s, 1H), 2.49 (s, 3H).

*6-(3-(Pyrimidin-4-yl)-1H-pyrazol-4-yl)quinoxaline* (**23**): Yield 58%; ^1^H NMR (300 MHz, DMSO-*d*_6_) *δ* 13.72 (s, 1H), 9.02 (s, 1H), 8.93 (br s, 2H), 8.85 (br s, 1H), 8.36 (br s, 1H), 8.14 (s, 1H), 8.03 (br s, 1H), 7.91 (d, *J* = 9.0 Hz, 2H).

#### 3.1.4. General Procedure for the Preparation of 2-(3-(6-(Dimethylamino)pyridin-2-yl)-, 2-(3-(4-Methylthiazol-2-yl)- or 2-(3-(Pyrimidin-4-yl)-4-(quinoxalin-6-yl)-1*H*-pyrazol-1-yl)acetamide (**14a**–**d**, **16a**–**d**, **25a**, **25b**, **25d**) and 2-(5-(6-(Dimethylamino)pyridin-2-yl)-, 2-(5-(4-Methylthiazol-2-yl)- or 2-(5-(Pyrimidin-4-yl)-4-(quinoxalin-6-yl)-1*H*-pyrazol-1-yl)-*N*-phenylacetamide (**15a**–**d**, **17a**–**d**, **26a**, **26b**, **26d**)

To a solution of pyrazole **12a**, **12b**, or **23** (0.63 mmol) in anhydrous DMF (8.3 mL), a catalytic amount of sodium iodide, NaH (0.75 mmol), and 2-chloro-*N*-phenylacetamide **13a**, **13b**, **13c**, **13d**, **24a**, **24b**, or **24d** (0.79 mmol) were added. The mixture was stirred at room temperature for 2 h and then evaporated to dryness under reduced pressure. The residue was purified by silica gel column chromatography (dichloromethane/methanol, 50:1) to give the two positional isomers **14a**–**d**, **16a**–**d**, **25a**, **25b**, **25d** and **15a**–**d**, **17a**–**d**, **26a**, **26b**, **26d** as white solids.

*2-(3-(6-(Dimethylamino)pyridin-2-yl)-4-(quinoxalin-6-yl)-1H-pyrazol-1-yl)-N-phenylacetamide* (**14a**): Yield 54%; HPLC purity: 98.68% (acetonitrile: 40%); mp 212.5–214.0 °C; ^1^H NMR (300 MHz, CDCl_3_) *δ* 8.87 (br s, 1H, NH), 8.84 (d, *J* = 6.0 Hz, 2H), 8.18 (s, 1H), 8.04 (d, *J* = 9.0 Hz, 1H), 7.84 (dd, *J* = 9.0, 3.0 Hz, 1H), 7.80 (s, 1H), 7.58–7.52 (m, 3H), 7.34 (t, *J* = 9.0 Hz, 2H), 7.15 (d, *J* = 9.0 Hz, 2H), 6.49 (d, *J* = 6.0 Hz, 1H), 5.06 (s, 2H), 2.72 (s, 6H); ^13^C NMR (75 MHz, CDCl_3_) *δ* 164.60, 158.64, 151.02, 149.38, 145.10, 144.50, 142.95, 142.02, 137.87, 137.07, 135.58, 132.76, 132.50, 129.02 (2C), 128.50, 128.14, 124.86, 121.69, 120.19 (2C), 109.86, 105.36, 55.91, 37.58 (2C); HRMS-ESI (*m*/*z*): [M + H]^+^ calcd for C_26_H_24_N_7_O 450.20368, found 450.20370.

*2-(3-(6-(Dimethylamino)pyridin-2-yl)-4-(quinoxalin-6-yl)-1H-pyrazol-1-yl)-N-(2-fluorophenyl)acetamide* (**14b**): Yield 40%; mp 190.5–193.0 °C; HPLC purity: 99.39% (acetonitrile: 40%); ^1^H NMR (300 MHz, CDCl_3_) *δ* 9.30 (s, 1H), 8.81 (d, *J* = 6.0 Hz, 2H), 8.31 (t, *J* = 7.5 Hz, 1H), 8.16 (d, *J* = 3.0 Hz, 1H), 8.01 (d, *J* = 9.0 Hz, 1H), 7.83 (dd, *J* = 9.0, 3.0 Hz, 1H), 7.76 (s, 1H), 7.54 (t, *J* = 7.5 Hz, 1H), 7.25 (d, *J* = 6.0 Hz, 1H), 7.17–7.06 (m, 3H), 6.45 (d, *J* = 9.0 Hz, 1H), 5.06 (s, 2H), 2.64 (s, 6H); ^13^C NMR (125 MHz, DMSO-*d*_6_) *δ* 166.30, 158.69, 153.95 (d, *J* = 243.75. Hz), 150.70, 149.00, 146.16, 145.38, 142.74, 141.50, 138.21, 136.43, 133.76, 132.70, 128.58, 127.50, 126.10 (d, *J* = 16.25 Hz), 126.09, 124.97 (d, *J* = 3.75 Hz), 124.24, 120.25, 116.07 (d, *J* = 18.75 Hz), 110.08, 105.21. 55.01, 30.48 (2C); HRMS-ESI (*m*/*z*): [M + H]^+^ calcd for C_26_H_23_FN_7_O 468.19426, found 468.19427.

*2-(3-(6-(Dimethylamino)pyridin-2-yl)-4-(quinoxalin-6-yl)-1H-pyrazol-1-yl)-N-(3-fluorophenyl)acetamide* (**14c**): Yield 50%; mp 176.5–178.5 °C; HPLC purity: 98.18% (acetonitrile: 40%); ^1^H NMR (300 MHz, CDCl_3_) *δ* 9.16 (s, 1H), 8.87–8.82 (m, 2H), 8.15 (s, 1H), 8.03 (d, *J* = 9.0 Hz, 1H), 7.83–7.78 (m, 2H), 7.56–7.48 (m, 2H), 7.32–7.22 (m, 2H), 7.16 (d, *J* = 9.0 Hz, 1H), 7.11 (d, *J* = 9.0 Hz, 1H), 6.83 (t, *J* = 8.0 Hz, 1H), 6.48 (d, *J* = 9.0 Hz, 1H), 5.03 (s, 2H), 2.71 (s, 6H); ^13^C NMR (75 MHz, CDCl_3_) *δ* 164.71, 161.23, 154.77 (d, *J* = 252.0 Hz), 153.97, 153.51, 145.10, 144.52, 142.90, 141.98, 138.28, 135.79, 135.36, 132.52 (d, *J* = 7.5 Hz), 130.03 (d, *J* = 9.0 Hz), 128.62, 128.04, 121.69, 115.44, 111.41 (d, *J* = 20.7 Hz), 110.07, 107.71, 107.40 (d, *J* = 6.75 Hz), 105.83, 55.82, 30.93 (2C); HRMS-ESI (*m*/*z*): [M + H]^+^ calcd for C_26_H_23_FN_7_O 468.19426, found 468.19431.

*N-(3-Cyanophenyl)-2-(3-(6-(dimethylamino)pyridin-2-yl)-4-(quinoxalin-6-yl)-1H-pyrazol-1-yl)acetamide* (**14d**): Yield 45%; mp 218.5–221.5 °C; HPLC purity: 99.52% (acetonitrile: 40%); ^1^H NMR (300 MHz, CDCl_3_) *δ* 9.39 (s, 1H), 8.83 (d, *J* = 3.0 Hz, 2H), 8.14 (d, *J* = 3.0 Hz, 1H), 8.02 (d, *J* = 9.0 Hz, 1H), 7.94 (s, 1H), 7.81–7.78 (m, 2H), 7.71–7.68 (m, 1H), 7.53 (t, *J* = 9.0 Hz, 1H), 7.41 (d, *J* = 6.0 Hz, 2H), 7.07 (d, *J* = 9.0 Hz, 1H), 6.49 (d, *J* = 9.0 Hz, 1H), 5.05 (s, 2H), 2.73 (s, 6H); ^13^C NMR (75 MHz, DMSO-*d*_6_) *δ* 166.48, 158.67, 150.65, 149.04, 146.17, 145.38, 142.72, 141.49, 139.80, 138.21, 136.41, 133.77, 132.69, 130.92, 128.59, 127.79, 127.50, 124.32, 122.42, 120.30, 119.07, 112.20, 110.08, 105.21, 55.28, 37.61 (2C); HRMS-ESI (*m*/*z*): [M + H]^+^ calcd for C_27_H_23_N_8_O 475.19893, found 475.19891.

*2-(3-(4-Methylthiazol-2-yl)-4-(quinoxalin-6-yl)-1H-pyrazol-1-yl)-N-phenylacetamide* (**16a**): Yield 65%; mp 190.2–192.5 °C; HPLC purity: 99.29% (acetonitrile: 45%); ^1^H NMR (300 MHz, CDCl_3_/DMSO-*d*_6_) *δ* 9.28 (s, 1H), 8.86 (br s, 2H), 8.27 (s, 1H), 8.10 (d, *J* = 9.0 Hz, 1H), 7.94 (d, *J* = 9.0 Hz, 1H), 7.88 (s, 1H), 7.58 (d, *J* = 9.0 Hz, 2H), 7.30 (d, *J* = 9.0 Hz, 2H), 7.10 (t, *J* = 6.0 Hz, 1H), 6.89 (s, 1H), 5.19 (s, 2H), 2.44 (s, 3H); ^13^C NMR (75 MHz, CDCl_3_) *δ* 164.16, 159.08, 153.48, 145.30, 144.90, 144.50 (2C), 142.95, 142.47, 137.05, 133.28, 132.86, 132.05, 129.06, 129.03 (2C), 124.92, 121.60, 120.14 (2C), 114.49, 55.99, 17.05; HRMS-ESI (*m*/*z*): [M + H]^+^ calcd for C_23_H_19_N_6_OS 427.13356, found 427.13354.

*N-(2-Fluorophenyl)-2-(3-(4-methylthiazol-2-yl)-4-(quinoxalin-6-yl)-1H-pyrazol-1-yl)acetamide* (**16b**): Yield 70%; mp 197.5–200.0 °C; HPLC purity: 96.75% (acetonitrile: 45%); ^1^H NMR (300 MHz, CDCl_3_) *δ* 8.89 (s,1H) 8.85 (d, *J* = 6.0 Hz, 2H), 8.32 (s, 1H), 8.28 (d, *J* = 9.0 Hz, 1H), 8.11 (d, *J* = 9.0 Hz, 1H), 8.04 (d, *J* = 9.0 Hz, 1H), 7.85 (s, 1H), 7.15–7.06 (m, 3H), 6.92 (s, 1H), 5.09 (s, 2H), 2.43 (s, 3H); ^13^C NMR (75 MHz, CDCl_3_) *δ* 164.07, 158.99, 153.61, 152.71 (d, *J* = 243.75 Hz), 145.17, 145.07, 144.79, 142.91, 142.45, 133.39, 132.77, 132.22, 128.95, 128.89, 128.76, 125.55 (d, *J* = 9.8 Hz), 125.21 (d, *J* = 7.5 Hz), 124.56 (d, *J* = 3.8 Hz), 121.80, 121.67, 115.02 (d, *J* = 19.5 Hz), 55.97, 17.12; HRMS-ESI (*m*/*z*): [M + H]^+^ calcd for C_23_H_18_FN_6_OS 445.12413, found 445.12405.

*N-(3-Fluorophenyl)-2-(3-(4-methylthiazol-2-yl)-4-(quinoxalin-6-yl)-1H-pyrazol-1-yl)acetamide* (**16c**): Yield 81%; mp 212.1–213.5 °C; HPLC purity: 100.00% (acetonitrile: 45%); ^1^H NMR (300 MHz, CDCl_3_) *δ* 9.41 (s, 1H), 8.85 (br s, 2H), 8.27 (s, 1H), 8.10 (d, *J* = 9.0 Hz, 1H), 8.02 (s, 1H), 7.95 (d, *J* = 9.0 Hz, 1H), 7.87 (s, 1H), 7.52 (d, *J* = 9.0 Hz, 1H), 7.23 (d, *J* = 9.0 Hz, 1H), 6.90 (s, 1H), 6.81 (d, *J* = 6.0 Hz, 1H), 5.16 (s, 2H), 2.43 (s, 3H); ^13^C NMR (75 MHz, CDCl_3_/CD_3_OD) *δ* 168.83, 167.13, 166.78 (d, *J* = 243.0 Hz), 163.54, 156.91, 149.04, 148.52, 147.19, 146.58, 145.91, 143.17 (d, *J* = 9.7 Hz), 138.13, 136.26, 133.97 (d, *J* = 9.5 Hz), 132.51, 132.16, 125.03, 119.20, 118.66, 115.09 (d, *J* = 21.3 Hz), 111.24 (d, *J* = 27.0 Hz), 59.22, 20.44; HRMS-ESI (*m*/*z*): [M + H]^+^ calcd for C_23_H_18_FN_6_OS 445.12413, found 445.12415.

*N-(3-Cyanophenyl)-2-(3-(4-methylthiazol-2-yl)-4-(quinoxalin-6-yl)-1H-pyrazol-1-yl)acetamide* (**16d**): Yield 74%; mp 222.3–223.0 °C; HPLC purity: 95.14% (acetonitrile: 45%); ^1^H NMR (300 MHz, CDCl_3_/DMSO-*d*_6_) *δ* 10.48 (s, 1H), 8.76 (d, *J* = 6.0 Hz, 2H), 8.28 (s, 1H), 8.04–7.95 (m, 4H), 7.75 (d, *J* = 9.0 Hz, 1H), 7.38 (d, *J* = 6.0 Hz, 1H), 7.32 (t, *J* = 6.0 Hz, 1H), 6.84 (s, 1H), 5.08 (s, 2H), 2.32 (s, 3H); ^13^C NMR (75 MHz, DMSO-*d*_6_) *δ* 166.11, 160.40, 152.90, 146.42, 145.74, 142.95, 142.79, 141.84, 139.73, 134.79, 134.25, 132.08, 130.93, 128.87, 128.11, 127.82, 124.32, 122.44, 119.87, 119.04, 115.33, 112.21, 55.37, 17.36; HRMS-ESI (*m*/*z*): [M + H]^+^ calcd for C_24_H_18_N_7_OS 452.12881, found 452.12878.

*N-Phenyl-2-(3-(pyrimidin-4-yl)-4-(quinoxalin-6-yl)-1H-pyrazol-1-**yl)acetamide* (**25a**): Yield 61%; mp 170.3–172.5 °C; HPLC purity: 99.97% (acetonitrile: 30%); ^1^H NMR (300 MHz, CDCl_3_) *δ* 9.14 (s, 1H), 8.85 (s, 2H), 8.71 (d, *J* = 6.0 Hz, 1H), 8.65 (s, 1H), 8.13 (d, *J* = 3.0 Hz, 1H), 8.08 (d, *J* = 6.0 Hz, 1H), 7.86 (s, 1H), 7.78 (dd, *J* = 9.0, 3.0 Hz, 1H), 7.67 (d, *J* = 6.0 Hz, 1H),7.48(d, *J* = 9.0 Hz, 2H), 7.30 (d, *J* = 6.0 Hz, 1H), 7.11 (t, *J* = 9.0 Hz, 1H), 5.12 (s, 2H); ^13^C NMR (125 MHz, CDCl_3_) *δ* 163.83, 159.00, 158.62, 157.47, 147.42, 145.43, 145.05, 143.00, 142.46, 136.79, 133.78, 133.19, 131.97, 129.31, 129.14 (2C), 128.86, 125.19, 123.29, 120.22 (2C), 119.02, 56.24. HRMS-ESI (*m*/*z*): [M + H]+ calcd for C_23_H_18_N_7_O 408.15673, found 408.15622.

*N-(2-Fluorophenyl)-2-(3-(pyrimidin-4-yl)-4-(quinoxalin-6-yl)-1H-pyrazol-1-yl)acetamide* (**25b**): Yield 58%; mp 192.3–194.5 °C; HPLC purity: 99.97% (acetonitrile: 30%); ^1^H NMR (300 MHz, CDCl_3_) *δ* 9.10 (s, 1H), 9.02 (br s, 1H, NH), 8.84 (s, 2H), 8.77 (d, *J* = 6.0 Hz, 1H), 8.29 (t, *J* = 7.5 Hz, 1H), 8.15 (d, *J* = 3.0 Hz, 1H), 8.10 (d, *J* = 6.0 Hz, 1H), 7.85–7.77 (m, 3H), 7.16–7.05 (m, 3H), 5.13 (s, 2H); ^13^C NMR (125 MHz, CDCl_3_) *δ* 163.72, 158.86, 158.57, 157.19, 152.56 (d, *J* = 244.0 Hz), 147.57, 145.27, 144.95, 142.90, 142.46, 133.88, 133.31, 132.24, 129.07, 128.83, 125.55 (d, *J* = 10.2 Hz), 125.22 (d, *J* = 7.7 Hz), 124.73 (d, *J* = 3.6 Hz), 123.45, 121.60, 118.80, 114.99 (d, *J* = 19.0 Hz), 56.13; HRMS (ESI) *m*/*z* calcd for C_23_H_17_FN_7_O 426.14731, found 426.14719.

*N-(3-Cyanophenyl)-2-(3-(pyrimidin-4-yl)-4-(quinoxalin-6-yl)-1H-pyrazol-**1-yl)acetamide* (**25d**): Yield 65%; mp 178.3–180.5 °C; HPLC purity: 99.27% (acetonitrile: 30%); ^1^H NMR (300 MHz, DMSO-*d*_6_) *δ* 10.86 (s, 1H), 9.03 (s, 1H), 8.95–8.93 (m, 2H), 8.85 (d, *J* = 6.0 Hz, 1H), 8.39 (s, 1H), 8.11 (s, 2H), 8.06 (d, *J* = 9.0 Hz, 1H), 7.91–7.88 (m, 3H), 7.59–7.58 (m, 2H), 5.28 (s, 2H); ^13^C NMR (125 MHz, DMSO-*d*_6_/CD_3_OD) *δ* 165.93, 159.82, 157.96, 157.08, 145.76, 145.46, 144.96, 142.54, 141.84, 139.25, 135.22, 134.16, 132.45, 130.03, 128.20, 127.82, 127.49, 123.96, 122.55, 122.17, 119.09, 118.18, 112.55, 54.91; HRMS-ESI (*m*/*z*): [M + H]+ calcd for C_24_H_17_N_8_O 433.15198, found 433.15179.

*2-(5-(6-(Dimethylamino)pyridin-2-yl)-4-(quinoxalin-6-yl)-1H-pyrazol-1-yl)-N-phenylacetamide* (**15a**): Yield 10%; 145.4–147.5 °C; ^1^H NMR (300 MHz, CDCl_3_) *δ* 8.81 (d, *J* = 6.0 Hz, 2H), 8.22 (s, 1H), 8.09 (s, 1H), 8.02–7.99 (m, 2H), 7.69 (dd, *J* = 9.0, 3.0 Hz, 1H), 7.49 (d, *J* = 9.0 Hz, 2H), 7.34–7.28 (m, 3H), 7.08 (t, *J* = 9.0 Hz, 1H), 6.52 (d, *J* = 9.0 Hz, 1H), 6.45 (d, *J* = 6.0 Hz, 1H), 5.17 (s, 2H), 3.07 (s, 6H); HRMS-ESI (*m*/*z*): [M + H]^+^ calcd for C_26_H_24_N_7_O 450.20368, found 450.20364.

*2-(5-(6-(Dimethylamino)pyridin-2-yl)-4-(quinoxalin-6-yl)-1H-pyrazol-1-yl)-N-(2-fluorophenyl)acetamide* (**15b**): Yield 13%; mp 130.2–133.0 °C; ^1^H NMR (300 MHz, CDCl_3_) *δ* 8.81 (d, *J* = 6.0 Hz, 2H), 8.45 (s, 1H), 8.30 (t, *J* = 7.5 Hz, 1H), 8.09 (s, 1H), 8.01–7.98 (m, 2H), 7.67 (dd, *J* = 9.0, 3.0 Hz, 1H), 7.33 (t, *J* = 9.0 Hz, 1H), 7.14–6.95 (m, 3H), 6.51 (d, *J* = 6.0 Hz, 1H), 5.24 (s, 2H), 3.07 (s, 6H); HRMS-ESI (*m*/*z*): [M + H]^+^ calcd for C_26_H_23_FN_7_O 468.19426, found 468.19406.

*2-(5-(6-(Dimethylamino)pyridin-2-yl)-4-(quinoxalin-6-yl)-1H-pyrazol-1-yl)-N-(3-fluorophenyl)acetamide* (**15c**): Yield 15%; mp 146.5–147.6 °C; ^1^H NMR (300 MHz, CDCl_3_) *δ* 8.81 (d, *J* = 3.0 Hz, 2H), 8.41 (s, 1H), 8.09 (s, 1H), 8.00 (d, *J* = 9.0 Hz, 2H), 7.68 (d, *J* = 9.0, 3.0 Hz, 1H), 7.50 (d, *J* = 9.0 Hz, 1H), 7.35 (t, *J* = 7.5 Hz, 1H), 7.23 (t, *J* = 9.0 Hz, 1H), 7.11 (d, *J* = 9.0 Hz, 1H), 6.81 (t, *J* = 9.0 Hz, 1H), 6.53 (d, *J* = 9.0 Hz, 1H), 6.45 (d, *J* = 6.0 Hz, 1H), 5.16 (s, 2H), 3.07 (s, 6H); HRMS-ESI (*m*/*z*): [M + H]^+^ calcd for C_26_H_23_FN_7_O 468.19426, found 468.19427.

*N-(3-Cyanophenyl)-2-(5-(6-(dimethylamino)pyridin-2-yl)-4-(quinoxalin-6-yl)-1H-pyrazol-1-yl)acetamide* (**15d**): Yield 14%; mp 145.5–148.0 °C; ^1^H NMR (300 MHz, CDCl_3_) *δ* 8.81 (d, *J* = 3.0 Hz, 2H), 8.63 (s, 1H), 8.08 (d, *J* = 3.0 Hz, 1H), 8.02–7.96 (m, 3H), 7.67 (d, *J* = 9.0 Hz, 2H), 7.43–7.33 (m, 3H), 6.54 (d, *J* = 9.0 Hz, 1H), 6.43 (d, *J* = 9.0 Hz, 1H), 5.18 (s, 2H), 3.07 (s, 6H); HRMS-ESI (*m*/*z*): [M + H]^+^ calcd for C_27_H_23_N_8_O 475.19893, found 475.19891.

*2-(5-(4-Methylthiazol-2-yl)-4-(quinoxalin-6-yl)-1H-pyrazol-1-yl)-N-phenylacetamide* (**17a**): Yield 8%; ^1^H NMR (300 MHz, CDCl_3_) *δ* 9.64 (s, 1H), 8.90 (s, 2H), 8.14 (d, *J* = 6.0 Hz, 1H), 8.04 (s, 4H), 7.83–7.65 (m, 1H), 7.60 (d, *J* = 6.0 Hz, 1H), 7.39–7.27 (m, 2H), 7.14 (d, *J* = 6.0 Hz, 1H), 5.29 (s, 2H), 2.60 (s, 3H); HRMS-ESI (*m*/*z*): [M + H]^+^ calcd for C_23_H_19_N_6_OS 427.13356, found 427.13361.

*N-(2-Fluorophenyl)-2-(5-(4-methylthiazol-2-yl)-4-(quinoxalin-6-yl)-1H-pyrazol-1-yl)acetamide* (**17b**): Yield 7%; ^1^H NMR (300 MHz, CDCl_3_) *δ* 9.66 (s, 1H), 8.86 (br s, 2H), 8.37 (t, *J* = 9.0 Hz, 1H), 8.13–8.09 (m, 2H), 7.87 (s, 1H), 7.73 (dd, *J* = 9.0, 3.0Hz, 1H), 7.16–7.03 (m, 3H), 6.98 (s, 1H), 5.33 (s, 2H), 2.57 (s, 3H); HRMS-ESI (*m*/*z*): [M + H]^+^ calcd for C_23_H_18_FN_6_OS 445.12413, found 445.12418.

*N-(3-Fluorophenyl)-2-(5-(4-methylthiazol-2-yl)-4-(quinoxalin-6-yl)-1H-pyrazol-1-yl)acetamide* (**17c**): Yield 7%; ^1^H NMR (300 MHz, CDCl_3_) *δ* 9.98 (s, 1H), 8.86 (br s, 2H), 8.14 (d, *J* = 9.0 Hz, 1H), 8.01 (br s, 2H), 7.87 (s, 1H), 7.72 (d, *J* = 9.0 Hz, 1H), 7.55 (d, *J* = 9.0 Hz, 1H), 7.25 (br s, 1H), 7.03 (br s, 1H), 6.81 (t, *J* = 9.0 Hz, 1H), 5.22 (s, 2H), 2.59 (s, 3H); HRMS-ESI (*m*/*z*): [M + H]^+^ calcd for C_23_H_18_FN_6_OS 445.12413, found 445.12408.

*N-(3-Cyanophenyl)-2-(5-(4-methylthiazol-2-yl)-4-(quinoxalin-6-yl)-1H-pyrazol-1-yl)acetamide* (**17d**): Yield 9%; ^1^H NMR (300 MHz, CDCl_3_) *δ* 10.43 (s, 1H), 8.75 (br s, 2H), 8.36 (d, *J* = 3.0 Hz, 1H), 8.26 (d, *J* = 9.0, 1H), 8.06–7.91 (m, 3H), 7.74–7.6 (m, 2H), 7.37–7.26 (m, *2*H), 5.42 (s, 2H), 2.34 (s, 3H); HRMS-ESI (*m*/*z*): [M + H]^+^ calcd for C_24_H_18_N_7_OS 452.12881, found 452.12881.

*N-Phenyl-2-(5-(pyrimidin-4-yl)-4-(quinoxalin-6-yl)-2,3-dihydro-1H-pyrazol-1-yl)acetamide* (**26a**): Yield 10%; HPLC purity: 96.43% (acetonitrile: 30%); ^1^H NMR (300 MHz, CDCl_3_) *δ* 9.43 (s, 1H), 8.87 (s, 2H), 8.80 (s, 1H), 8.66 (d, *J* = 6.0 Hz, 1H), 8.11 (d, *J* = 9.0 Hz, 1H), 8.05 (s, 1H), 7.95 (s, 1H), 7.62 (dd, *J* = 9.0, 3.0 Hz, 1H), 7.53 (d, *J* = 6.0 Hz, 2H), 7.33 (t, *J* = 7.5 Hz, 2H), 7.20 (dd, *J* = 6.0, 3.0 Hz, 1H), 7.13 (t, *J* = 7.5 Hz, 1H), 5.29 (s, 2H); HRMS-ESI (*m*/*z*): [M + H]+ calcd for C_23_H_18_N_7_O 408.15673, found 408.15698.

*N-(2-Fluorophenyl)-2-(5-(pyrimidin-4-yl)-4-(quinoxalin-6-yl)-2,3-dihydro-1H-pyrazol-1-yl)acetamide* (**26b**): Yield 7%; HPLC purity: 99.29% (acetonitrile: 30%); ^1^H NMR (300 MHz, CDCl_3_) *δ* 9.55 (s, 1H), 9.45 (s, 1H), 8.86 (s, 2H), 8.65 (d, *J* = 6.0 Hz, 1H), 8.34 (t, *J* = 9.0 Hz, 1H), 8.11 (d, *J* = 9.0 Hz, 1H), 8.06 (s, 1H), 7.94 (s, 1H), 7.62 (dd, *J* = 9.0, 3.0 Hz, 1H), 7.19 (d, J = 6.0 Hz, 1H), 7.15–7.05 (m, 3H), 5.31 (s, 2H); HRMS-ESI (*m*/*z*): [M + H]+ calcd for C_23_H_17_FN_7_O 426.14731, found 426.14783.

*N-(3-Cyanophenyl)-2-(5-(pyrimidin-4-yl)-4-(quinoxalin-6-yl)-2,3-dihydro-1H-pyrazol-1-yl)acetamide (***26d**): Yield 13%; HPLC purity: 97.98% (acetonitrile: 45%); ^1^H NMR (300 MHz, CDCl_3_) *δ* 10.04 (s, 1H), 9.25 (s, 1H), 8.78 (br s, 2H), 8.54(s, 1H), 7.96 (br s, 3H), 7.89–7.81 (m, 2H), 7.69 (br s, 1H), 7.58 (br s, 1H), 7.26 (s, 1H), 7.19 (s, 1H), 5.44 (s, 2H); HRMS (ESI) *m*/*z* calcd for C_24_H_17_N_8_O 433.15198, found 433.15216.

#### 3.1.5. General Procedure for the Preparation of 2-(3-(6-(Dimethylamino)pyridin-2-yl)-4-(quinoxalin-6-yl)-1*H*-pyrazol-1-yl)-*N*-phenylethanethioamide **18a**–**d**, 2-(3-(4-Methylthiazol-2-yl)-4-(quinoxalin-6-yl)-1*H*-pyrazol-1-yl)-*N*-phenylethanethioamide **19a**–**d** or *N*-Phenyl-2-(3-(pyrimidin-4-yl)-4-(quinoxalin-6-yl)-1*H*-pyrazol-1-yl)ethanethioamide (**27b and 27d**)

A stirred mixture of **14a**–**d**, **16a**–**d**, **25b**, or **25d** (0.34 mmol), Lawesson’s reagent (0.34 mmol), and anhydrous DME (5 mL) in a dry sealed tube was heated at 85 °C for 12 h. After cooled to room temperature, the solvent was evaporated to dryness under reduced pressure, and the residue was purified by silica gel column chromatography (dichloromethane/methanol, 100:1) to give the titled compounds **18a**–**d**, **19a**–**d**, **27b**, or **27d** as a light yellow solid.

*2-(3-(6-(Dimethylamino)pyridin-2-yl)-4-(quinoxalin-6-yl)-1H-pyrazol-1-yl)-N-phenylethanethioamide* (**18a**): Yield 45%; mp 182.0–184.0 °C; HPLC purity: 96.95% (acetonitrile: 40%); ^1^H NMR (300 MHz, CDCl_3_) *δ* 10.79 (s, 1H), 8.82 (d, *J* = 6.0 Hz, 2H), 8.15 (s, 1H), 8.03 (d, *J* = 9.0 Hz, 1H), 7.82–7.70 (m, 3H), 7.52 (t, *J* = 7.5 Hz, 2H), 7.39 (t, *J* = 7.5 Hz, 2H), 7.26 (d, *J* = 9.0 Hz, 1H), 7.09 (d, *J* = 9.0 Hz, 1H), 6.49 (d, *J* = 9.0 Hz, 1H), 5.44 (s, 2H), 2.74 (s, 6H); ^13^C NMR (75 MHz, CDCl_3_) *δ* 192.76, 167.73, 145.14, 144.56, 142.96, 142.06, 138.24, 138.00, 132.68, 132.39, 132.31, 130.93 (2C), 128.90, 128.85 (2C), 128.60, 128.25, 127.00, 123.04, 121.71, 109.87, 105.56, 65.58, 37.67 (2C); HRMS-ESI (*m*/*z*): [M + H]^+^ calcd for C_26_H_24_N_7_S 466.18084, found 466.18082.

*2-(3-(6-(Dimethylamino)pyridin-2-yl)-4-(quinoxalin-6-yl)-1H-pyrazol-1-yl)-N-(2-fluorophenyl)ethanethioamide* (**18b**): Yield 43%; mp 88.3–91.2 °C; HPLC purity: 99.31% (acetonitrile: 40%); ^1^H NMR (300 MHz, CDCl_3_) *δ* 10.79 (s, 1H), 8.82 (d, *J* = 6.0 Hz, 2H), 8.70 (t, *J* = 7.5 Hz, 1H), 8.16 (d, *J* = 3.0 Hz, 1H), 8.01 (d, *J* = 9.0 Hz, 1H), 7.82 (dd, *J* = 9.0, 3.0 Hz, 1H), 7.79 (s, 1H), 7.53 (t, *J* = 7.5 Hz, 1H), 7.24–7.17 (m, 4H), 6.46 (d, *J* = 9.0 Hz, 1H), 5.44 (s, 2H), 2.64 (s, 6H); ^13^C NMR (125 MHz, CDCl_3_) *δ* 193.51, 158.67, 154.33 (d, *J* = 248.6 Hz), 151.59, 149.39, 145.06, 144.48, 142.98, 142.07, 137.80, 135.72, 132.94, 132.23, 128.36 (d, *J* = 18.0 Hz), 127.63 (d, *J* = 7.8 Hz), 126.72 (d, *J* = 10.0 Hz), 124.02 (d, *J* = 3.8 Hz), 123.79, 121.91, 115.48 (d, *J* = 19.1 Hz), 109.76, 105.30, 63.66, 37.48 (2C); HRMS-ESI (*m*/*z*): [M + H]^+^ calcd for C_26_H_23_FN_7_S 484.17142, found 484.17133.

*2-(3-(6-(Dimethylamino)pyridin-2-yl)-4-(quinoxalin-6-yl)-1H-pyrazol-1-yl)-N-(3-fluorophenyl)ethanethioamide* (**18c**): Yield 44%; mp 68.5–70.2 °C; HPLC purity: 98.45% (acetonitrile: 40%); ^1^H NMR (300 MHz, CDCl_3_) *δ* 10.94 (s, 1H), 8.82 (d, *J* = 6.0 Hz, 2H), 8.14 (d, *J* = 3.0 Hz, 1H), 8.02 (d, *J* = 9.0 Hz, 1H), 7.88 (d, *J* = 9.0 Hz, 1H), 7.81–7.78 (m, 2H), 7.52 (t, *J* = 7.5 Hz, 1H), 7.42 (d, *J* = 9.0 Hz, 1H), 7.34 (t, *J* = 9.0 Hz, 1H), 7.09 (d, *J* = 9.0 Hz, 1H), 6.96 (t, *J* = 9.0 Hz, 1H), 6.47 (d, *J* = 9.0 Hz, 1H), 5.40 (s, 2H), 2.71 (s, 6H); ^13^C NMR (75 MHz, CDCl_3_) *δ* 192.94, 164.11, 159.68 (d, *J* = 175.7 Hz), 148.85, 145.17, 144.61, 142.96, 142.08, 139.63 (d, *J* = 10.5 Hz), 138.03, 135.26, 133.85, 132.50 (d, *J* = 15.0 Hz), 130.06 (d, *J* = 9.2 Hz), 128.67, 128.27, 121.74, 118.38 (d, *J* = 3.2 Hz), 113.74 (d, *J* = 21.2 Hz), 110.24, 109.89, 105.66, 63.58, 37.69 (2C); HRMS-ESI (*m*/*z*): [M + H]^+^ calcd for C_26_H_23_FN_7_S 484.17142, found 484.17133.

*N-(3-Cyanophenyl)-2-(3-(6-(dimethylamino)pyridin-2-yl)-4-(quinoxalin-6-yl)-1H-pyrazol-1-yl)ethanethioamide* (**18d**): Yield 37%; mp 108.5–110.0 °C; HPLC purity: 96.46% (acetonitrile: 40%); ^1^H NMR (300 MHz, CDCl_3_) *δ* 11.13 (s, 1H), 8.83 (d, *J* = 6.0 Hz, 2H), 8.26 (s, 1H), 8.14 (s, 1H), 8.01 (t, *J* = 9.0 Hz, 2H), 7.81–7.78 (m, 2H), 7.54–7.49 (m, 3H), 7.05 (d, *J* = 9.0 Hz, 1H), 6.49 (d, *J* = 9.0 Hz, 1H), 5.41 (s, 2H), 2.73 (s, 6H); ^13^C NMR (75 MHz, DMSO-*d*_6_) *δ* 197.16, 158.56, 146.18, 145.39, 142.71, 141.48, 140.23, 138.33, 137.50, 136.35, 134.03, 132.69, 130.75, 130.50, 128.59, 127.49, 126.65, 120.36, 120.15, 118.77, 111.89, 110.23, 105.33, 62.82, 39.12 (2C); HRMS-ESI (*m*/*z*): [M + H]^+^ calcd for C_27_H_23_N_8_S 491.17609, found 491.17609.

*2-(3-(4-Methylthiazol-2-yl)-4-(quinoxalin-6-yl)-1H-pyrazol-1-yl)-N-phenylethanethioamide* (**19a**): Semi-solid; Yield 80%; HPLC purity: 96.56% (acetonitrile: 45%); ^1^H NMR (300 MHz, CDCl_3_) *δ* 11.02 (s, 1H), 8.91 (br s, 2H), 8.29 (s, 1H), 8.17 (t, *J* = 9.0 Hz, 1H), 7.96–7.88 (m, 3H), 7.38 (t, *J* = 9.0 Hz, 2H), 7.26 (b r s, 2H), 6.96 (s, 1H), 5.54 (s, 2H), 2.57 (s, 3H); ^13^C NMR (125 MHz, CDCl_3_) *δ* 192.07, 158.81, 153.81, 145.31, 145.10, 144.93, 142.98, 142.53, 138.17, 133.19, 132.62, 132.07, 129.16, 129.07, 128.99 (2C), 127.12, 122.94 (2C), 121.70, 114.54, 63.83, 17.13; HRMS-ESI (*m*/*z*): [M + H]^+^ calcd for C_23_H_19_N_6_S_2_ 443.11071, found 443.11072.

*N-(2-Fluorophenyl)-2-(3-(4-methylthiazol-2-yl)-4-(quinoxalin-6-yl)-1H-pyrazol-1-yl)ethanethioamide* (**19b**): Semi-solid; Yield 84%; HPLC purity: 96.40% (acetonitrile: 45%); ^1^H NMR (300 MHz, DMSO-*d*_6_) *δ* 11.79 (s, 1H), 8.94 (d, *J* = 9.0 Hz, 2H), 8.50 (d, *J* = 9.0 Hz, 2H), 8.16 (d, *J* = 9.0 Hz, 1H), 8.08 (d, *J* = 9.0 Hz, 1H), 7.73–7.61 (m, 4H), 7.39–7.25 (m, 2H), 5.48 (s, 2H), 2.35 (s, 3H); ^13^C NMR (75 MHz, CDCl_3_/DMSO-*d*_6_) *δ* 194.18, 167.96, 159.08, 154.96 (d, *J* = 248.25 Hz), 153.40, 145.09, 144.63, 142.66, 142.09, 133.79, 132.61, 132.27, 131.08 (d, *J* = 14.4 Hz), 130.99, 130.43, 128.77 (d, *J* = 3.0 Hz), 128.60 (d, *J* = 8.3 Hz), 128.21 (d, *J* = 8.3 Hz), 125.28, 115.68 (d, *J* = 19.4 Hz), 114.68, 65.63, 19.07; HRMS-ESI (*m*/*z*): [M + H]^+^ calcd for C_23_H_18_FN_6_S_2_ 461.10129, found 461.10120.

*N-(3-Fluorophenyl)-2-(3-(4-methylthiazol-2-yl)-4-(quinoxalin-6-yl)-1H-pyrazol-1-yl)ethanethioamide* (**19c**): Semi-solid; Yield 88%; HPLC purity: 96.37% (acetonitrile: 45%); ^1^H NMR (300 MHz, DMSO-*d*_6_) *δ* 12.21 (s, 1H), 8.93 (d, *J* = 9.0 Hz, 2H), 8.52 (s, 1H), 8.48 (s, 1H), 8.16 (d, *J* = 9.0 Hz, 1H), 8.06 (t, *J* = 9.0 Hz, 1H), 7.73–7.68 (m, 2H), 7.50 (d, *J* = 9.0 Hz, 1H), 7.31 (s, 1H), 7.14 (s, 1H), 5.43 (s, 2H), 2.34 (s, 3H); ^13^C NMR (75 MHz, CDCl_3_/DMSO-*d*_6_) *δ* 197.82, 166.35 (d, *J* = 243.0 Hz), 163.66, 157.17, 149.06, 148.50, 147.52, 146.54, 145.81, 138.34, 136.12 (d, *J* = 17.5 Hz), 135.66, 135.10 (d, *J* = 11.4 Hz), 133.83 (d, *J* = 9.0 Hz), 132.68, 132.34, 124.62, 122.40 (d, *J* = 2.3 Hz), 118.65, 117.20 (d, *J* = 21.0 Hz), 114.12, 69.57, 17.24; HRMS-ESI (*m*/*z*): [M + H]^+^ calcd for C_23_H_18_FN_6_S_2_ 461.10129, found 461.10132.

*N-(3-Cyanophenyl)-2-(3-(4-methylthiazol-2-yl)-4-(quinoxalin-6-yl)-1H-pyrazol-1-yl)ethanethioamide* (**19d**): Semi-solid; Yield 89%; HPLC purity: 96.31% (acetonitrile: 45%); ^1^H NMR (300 MHz, DMSO-*d*_6_) *δ* 12.33 (s, 1H), 8.93 (d, *J* = 9.0 Hz, 2H), 8.50 (d, *J* = 6.0 Hz, 2H), 8.44 (s, 1H), 8.14–8.06 (m, 2H), 7.77 (d, *J* = 6.0 Hz, 1H), 7.69 (t, *J* = 7.5 Hz, 1H), 7.31 (s, 1H), 6.91 (s, 1H), 5.45 (s, 2H), 2.34 (s, 3H); ^13^C NMR (75 MHz, DMSO-*d*_6_) *δ* 196.70, 160.48, 152.89, 146.42, 145.73, 143.12, 142.80, 141.82, 140.22, 135.10, 134.31, 132.08, 130.75, 130.48, 128.87, 128.53, 128.08, 126.60, 119.71, 118.76, 115.34, 111.90, 61.84, 17.36; HRMS-ESI (*m*/*z*): [M + H]^+^ calcd for C_24_H_18_N_7_S_2_ 468.10596, found 468.10599.

*N-(2-Fluorophenyl)-2-(3-(pyrimidin-4-yl)-4-(quinoxalin-6-yl)-1H-pyrazol-1-yl)ethanethioamide* (**27b**): Yield 68%; HPLC purity: 98.75% (acetonitrile: 35%); ^1^H NMR (300 MHz, CDCl_3_) *δ* 10.57 (s, 1H, NH), 9.12 (s, 1H), 8.84 (s, 2H), 8.78 (s, 1H), 8.64 (t, *J* = 7.5 Hz, 1H), 8.16 (s, 1H), 8.08 (d, *J* = 9.0 Hz, 1H), 7.92 (s, 1H), 7.82 (d, *J* = 9.0 Hz, 2H), 7.23–7.10 (m, 3H), 5.48 (s, 2H); ^13^C NMR (125 MHz, CDCl_3_) *δ* 192.31, 167.73, 158.80, 157.49, 154.12 (d, *J* = 247.5 Hz), 147.81, 145.36, 145.01, 142.94, 142.47, 133.77, 133.11, 132.15, 130.92, 128.89, 128.85, 127.72 (d, *J* = 7.5 Hz), 126.62 (d, *J* = 10.0 Hz), 124.19 (d, *J* = 3.75 Hz), 123.47, 118.84, 115.44 (d, *J* = 18.75 Hz), 65.58; HRMS-ESI (*m*/*z*): [M + H]^+^ calcd for C_2__3_H_1__7_FN_7_S 442.12447, found 442.12447.

*N-(3-Cyanophenyl)-2-(3-(pyrimidin-4-yl)-4-(quinoxalin-6-yl)-1H-pyrazol-1-yl)ethanethioamide* (**27d**): Yield 50%; HPLC purity: 99.68% (acetonitrile: 35%); ^1^H NMR (300 MHz, CDCl_3_) *δ* 10.57 (s, 1H, NH), 9.21 (s, 1H), 8.88 (s, 2H), 8.76 (d, *J* = 6.0 Hz, 1H), 8.27 (s, 1H), 8.16–8.11 (m, 2H), 7.91 (br s, 2H), 7.79 (d, *J* = 9.0 Hz, 1H), 7.64 (s, 1H), 7.54–7.49 (m, 2H), 5.50 (s, 2H); ^13^C NMR (125 MHz, CDCl_3_) *δ* 194.60, 159.15, 158.61, 157.07, 146.27, 145.21, 144.73, 142.88, 142.19, 139.77, 134.36, 133.27, 132.07, 129.72, 129.61, 128.89, 128.47, 127.38, 126.19, 122.38, 119.06, 118.16, 112.49, 63.61; HRMS-ESI (*m*/*z*): [M + H]+ calcd for C_24_H_17_N_8_S 449.12914, found 449.1285.

### 3.2. Kinase Assay

All kinase experiments were completed by the ProQinase (Freiburg, Germany). All protein kinases were expressed in Sf9 insect cells or in *E. coli* as recombinant GST-fusion proteins or His-tagged proteins, either as full-length or enzymatically active fragments. All kinases were obtained from human cDNAs and purified by either GSH-affinity chromatography or immobilized metal. The purity of the protein kinases was examined by SDS-PAGE/Coomassie staining. The identity was checked by mass spectroscopy.

A radiometric protein kinase assay (^33^PanQinase^®^ activity assay) was used for measuring the kinase activity of the two protein kinases. All kinase assays were performed in 96-well FlashPlates^TM^ from PerkinElmer (Boston, MA, USA) in 50 μL reaction volumes. The reaction cocktail was pipetted in four steps in the following order: 20 μL of assay buffer (standard buffer), 5 μL of ATP solution (in H_2_O), 5 μL of test compound (in 10% DMSO), 20 μL enzyme/substrate mix.

The assay for all protein kinases contained 70 mM HEPES-NaOH pH 7.5, 3 mM MgCl2, 3 mM MnCl2, 3 μM Na-orthovanadate, 1.2 mM DTT, 50 μg/mL PEG_20000_, ATP, [γ-^33^P]-ATP, protein kinase, and substrate.

The reaction cocktail was incubated at 30 °C for 60 min. The reaction was halted with 50 μL of 2% (*v*/*v*) H_3_PO_4_, plates were aspirated and washed two times with 200 μL 0.9% (*w*/*v*) NaCl. Incorporation of ^33^Pi was established with a microplate scintillation counter (PerkinElmer, Boston, MA, USA). All assays were performed with a BeckmanCoulter/SAGIAN™ Core System.

### 3.3. Docking Assay

All molecular computation studies were carried out using Discovery Studio 2017 (Accelrys, San Diego, CA, USA). The X-ray crystal structure of ALK5 complexed with 5,6-dihydro-4H-pyrrolo[1,2-b]pyrazole inhibitor was obtained from protein data bank (PDB: 1RW8). The water molecules and heavy atom in protein were removed and the protein was prepared by adding hydrogen and correcting incomplete residues using Clean Protein tool of DS, then the protein was refined with CHARMm. The structures of compounds **18b** and **19b** were sketched in 2D and converted into 3D using the DS molecule editor. Automated docking studies were carried out to investigate the binding mode of compounds **18b** and **19b** in the crystal structure of 5,6-dihydro-4H-pyrrolo[1,2-b]pyrazole utilizing DS-CDOCKER protocol. The pose with the top CDOCKER_INTERACTION_ENERGY was chosen for analyzing the binding features of compounds **18b** and **19b** with ALK5.

### 3.4. Prediction of ADMET Properties

ADMET properties of good targeted compounds **19a**–**d** as drug lead compound were predicted using ADMET descriptors in Discovery Studio 2017 (Accelrys, San Diego, CA, USA). It is a quick, easy and accurate method for prediction of absorption, distribution, metabolism, elimination and toxicity (ADMET) properties. In this work, for the aforementioned compounds, human intestinal absorption level, aqueous solubility (log(SW)), blood–brain barrier (BBB) penetration level (AlogP98), human cytochrome P450 2D6 (CYP2D6) inhibitory ability, hepatotoxicity possibility, and plasma protein binding (PPB) levels were measured.

## 4. Conclusions

In our study, 32 quinoxaline-derivatives of 3-substituted-4-(quinoxalin-6-yl) pyrazoles **14a**–**d**, **15a**–**d**, **16a**–**d**, **17a**–**d**, **18a**–**d**, **19a**–**d**, **25a**, **25b**, **25d**, **26a**, **26b**, **26d**, **27b**, and **27d** were synthesized and evaluated for ALK5 and p38α MAP kinase inhibitory activities in enzymatic assays. We found that insertion of a 4-methylthiazol-2-yl moiety at the 3-position of the pyrazole ring showed more potent ALK5 inhibitory activity and selectivity than introduction of electron-donating groups into quinoxaline or quinoline ring. The most potent compound, **19b**, inhibited ALK5 phosphorylation with an IC_50_ value of 0.28 µM and showed 98% inhibition at 10 µM in the enzymatic assay. The selectivity index of **19b** against p38α MAP kinase was >35, much higher than that of positive control compound **3** (4). Although compound **19b** has slightly lower activity than the positive control compound **3**, its selectivity is much higher than that of the positive control compound **3**, so its side effects may be lower. The docking study described that compounds possessing 4-methylthazol-2-yl moiety was found to show better docking interaction than compounds possessing 6-(dimethylamino)pyridine-2-yl moiety on its active site. All good targeted compounds were subjected to ADMET prediction and the predicted ADMET parameters were within the acceptable range defined for human use. This result provides good data for the design of substituents for the 3-position introduction of the pyrazole ring, which may be a good choice if the lipophilic compounds are needed. In particular, compound **19b** was the most promising and it could be considered worthwhile lead compound worthy of further investigation.

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
