# Peer review of "Synthesis and Evaluation of 3-Substituted-4-(quinoxalin-6-yl) Pyrazoles as TGF-β Type I Receptor Kinase Inhibitors"

_molecules, 2018, doi:10.3390/molecules23123369_

Round 1

Reviewer 1 Report

the manuscript should be rewritten, to avoid overlaps in the synthesis of the compounds with the previous works:European Journal of Medicinal Chemistry 46 (2011) 3917-3925 and Bioorg. Med. Chem. 19 (2011) 2633–2, in order to highlight the innovation that is provided.

Author Response

Point : The manuscript should be rewritten, to avoid overlaps in the synthesis of the compounds with the previous works: European Journal of Medicinal Chemistry 46 (2011) 3917-3925 and Bioorg. Med. Chem. 19 (2011) 2633–2, in order to highlight the innovation that is provided.

Response: Thanks for your important suggestion. According to your suggestion, the manuscript has been greatly modified to avoid overlaps in the synthesis of the compounds with the previous works. Revised portions are marked in red in the paper. 

Reviewer 2 Report

The paper entitled "Synthesis and evaluation of 3-substituted-4-(quinoxalin-6-yl) pyrazoles as TGF-β type I receptor kinase inhibitors" by Li-Min Zhaot et al. presents the synthesis, the TGF-B type I receptor kinase inhibitory activity (enzymatic and molecular docking studies) and ADMET properties. Overall the research's design is appropiate and very interesting, BUT the description of the methods (especially the synthesis part) reduces a lot the quality of this work. Considering the number of the compounds (32, if I counted well, because it is very difficult to follow) a very logical numbering is crucial. For example, the structures of the compounds 22 (a,b,d) and 23 (a,b,d) is not at all clear, since there is only one subsituent (R) in their structure. Moreover, since the synthesis of all these compounds is based on compound 6 (replacement of thioamide linkage with a thioamide-methylene one) it would be logical to use this compound, instead of co. 3, as reference in the biological activity assays. Further, there are some Eglish language errors, that must be corrected.

And finally, did not 100% respect the editing demands of this journal (the text is not justified, there are a lot of spaces that are unnecessary). 

Therefore, I strongly advice the authors to rethink the whole numbering of these compounds, use compound 6 as reference, or at least justify the use of compound 3, and send the paper again, in a more understandable and easier to follow form.

Author Response

Dear editor and reviewers:

Thank you your letter and giving us an opportunity to revise our manuscript, we appreciate editor and reviewers very much for their positive and constructive comments and suggestions on our manuscript entitled “Synthesis and evaluation of 3-substituted-4-(quinoxalin-6-yl) pyrazoles as TGF-β type I receptor kinase inhibitors” (molecules-404024). Those positive and constructive comments are very helpful for revising and improving our paper, and they are also very important to our following research work.

We have read comments carefully and have tried our best to revise our manuscript according to the comments. Revised portion are marked in red in the paper. The main corrections in the paper and the responds to the reviewer’s comments are as flowing: 

Point 1: The paper entitled "Synthesis and evaluation of 3-substituted-4-(quinoxalin-6-yl) pyrazoles as TGF-β type I receptor kinase inhibitors" by Li-Min Zhaot et al. presents the synthesis, the TGF-B type I receptor kinase inhibitory activity (enzymatic and molecular docking studies) and ADMET properties. Overall the research's design is appropiate and very interesting, BUT the description of the methods (especially the synthesis part) reduces a lot the quality of this work. Considering the number of the compounds (32, if I counted well, because it is very difficult to follow) a very logical numbering is crucial. For example, the structures of the compounds 22 (a,b,d) and 23 (a,b,d) is not at all clear, since there is only one subsituent (R) in their structure. Moreover, since the synthesis of all these compounds is based on compound 6 (replacement of thioamide linkage with a thioamide-methylene one) it would be logical to use this compound, instead of co. 3, as reference in the biological activity assays. Further, there are some Eglish language errors, that must be corrected.

Response 1: Thanks for your important suggestion.

    In accordance with your advice, we have renumbered all the compounds in the manuscript.

    We are very sorry for the mistake. According to your advice, we have corrected the spelling, grammar, and punctuation throughout the manuscript.

Point 2: And finally, did not 100% respect the editing demands of this journal (the text is not justified, there are a lot of spaces that are unnecessary). 

Response 2: Thanks for your suggestions.

We are very sorry for the mistake. According to your suggestion, we have edited the paper as a whole according to the format of the paper.

Point 3: Therefore, I strongly advice the authors to rethink the whole numbering of these compounds, use compound 6 as reference, or at least justify the use of compound 3, and send the paper again, in a more understandable and easier to follow form.

Response 3: Thanks for your valuable advice.

    In accordance with your advice, we have renumbered all the compounds in the manuscript.

    At first, we tried compound 6 as a positive control compound, but the results were inconsistent due to its unstable structure and activity. We selected compound 3 as reference in the biological activity assays, because it has progressed to Phase II trials as antitumor agent. We believe that the synthesized compounds can be compared with the compound 3 to determine the activity and druggability of the series of compounds, and the positive control compound was used.  

Reviewer 3 Report

This manuscript mainly presents the structure-activity relationship of pyrazole derivatives for producing the TGF-b type-I receptor kinase (ALK5) inhibitors. The manuscript should be accepted but be necessary for major and minor revisions.

1) In Figure 3, the graphical items B and D should be modified. The characters for amino acid names are too small, thus authors should use large and conspicuous letters. Readers cannot imagine the orientation of the ATP site, thus authors should specify the hinge region residue, active lysine, and so on in these items.

2) In Figure 3 and line 212, authors give 1RWB for ALK5 structure but it is for glucose dehydrogenase. Authors should replace it with correct PDB ID. 1RW8?

3) In lines 37-38, the word attenuate is non-contextualized, thus authors should replace attenuate with result in.

4) In line 51, the authors suggested that the selectivity for p38 MAP kinase is considered as a serious problem but did not present that reason. Authors should insert the significance of the selectivity for p38 in this part as shown in line 164.

Author Response

Dear editor and reviewers:

Thank you your letter and giving us an opportunity to revise our manuscript, we appreciate editor and reviewers very much for their positive and constructive comments and suggestions on our manuscript entitled “Synthesis and evaluation of 3-substituted-4-(quinoxalin-6-yl) pyrazoles as TGF-β type I receptor kinase inhibitors” (molecules-404024). Those positive and constructive comments are very helpful for revising and improving our paper, and they are also very important to our following research work.

We have read comments carefully and have tried our best to revise our manuscript according to the comments. Revised portion are marked in red in the paper. The main corrections in the paper and the responds to the reviewer’s comments are as flowing: 

 Point 1: In Figure 3, the graphical items B and D should be modified. The characters for amino acid names are too small, thus authors should use large and conspicuous letters. Readers cannot imagine the orientation of the ATP site, thus authors should specify the hinge region residue, active lysine, and so on in these items.

Response 1: We are very sorry for the mistake. According to your suggestion, we enlarged the amino acid names of the graphical items B and D in figure 4 and adjusted the angle of the 3D diagram to observe the corresponding binding sites.

Point 2: In Figure 3 and line 212, authors give 1RWB for ALK5 structure but it is for glucose dehydrogenase. Authors should replace it with correct PDB ID. 1RW8?

 Response 2: We are very sorry for the mistake. We have corrected them in line 215 and in Figure in the manuscript.  

Point 3: In lines 37-38, the word “attenuate” is non-contextualized, thus authors should replace “attenuate” with “result in”.

Response 3: Thanks for your suggestion. We have corrected it in line 42 in the manuscript.

Point 4: In line 51, the authors suggested that the selectivity for p38 MAP kinase is considered as a serious problem but did not present that reason. Authors should insert the significance of the selectivity for p38 in this part as shown in line 164.

Response 4: Thanks for your valuable advice. Following your suggestion, we have corrected it in line 54 in the manuscript. At the beginning of the study, we hypothesized that if the compound was selective for kinases most similar to the target, it would also be selective for other kinases. As we expected, we found that compounds with high selectivity to p38α MAP kinase showed high selectivity to other kinases in the study. There, we took p38α MAP kinase as the index to evaluate ALK5 kinase selectivity. 

Round 2

Reviewer 1 Report

The authors have followed the suggestions adequately

Author Response

Dear editor and reviewers:

Thank you your letter and giving us an opportunity to revise our manuscript, we appreciate editor and reviewers very much for their positive and constructive comments and suggestions on our manuscript entitled “Synthesis and evaluation of 3-substituted-4-(quinoxalin-6-yl) pyrazoles as TGF-β type I receptor kinase inhibitors” (molecules-404024). Those positive and constructive comments are very helpful for revising and improving our paper, and they are also very important to our following research work.

We have read comments carefully and have tried our best to revise our manuscript according to the comments. Revised portion are marked in red in the paper. The main corrections in the paper and the responds to the reviewer’s comments are as flowing:

Point 1: The authors have followed the suggestions adequately.

Response: Thank you for your affirmation and suggestion.

Reviewer 2 Report

The paper entitled "Synthesis and evaluation of 3-substituted-4-(quinoxalin-6-yl) pyrazoles as TGF-β type I receptor kinase inhibitors" by Li-Min Zhaot et al. presents the synthesis, the TGF-B type I receptor kinase inhibitory activity (enzymatic and molecular docking studies) and ADMET properties. As I noticed, the paper suffered major modifications, but it still needs to be improved before acceptance. Here are my suggestions:

in the Abstract, but also in the Conclusion, replace "three series" with 32 quinoxaline-derivatives, since you mention 9 series in both parts;

reformulate the phrase of lines 60-62;

lines 68-70: What's the link to all mentioned before? Delete this part or explain it while linking it to the above mentioned;

line 76: replace "assume" with "assumed";

Scheme 1: insert the structures of compounds 8 and 9;

line 127: replace "diemthyl" with "dimethyl";

line 145: delete "strong";

lines 148 and 150: replace "inhibition activity" with "inhibitory activity"

line 158: replace "capability" with "capacity";

remake all the References part, since most of the articles cited are more than 10 years old and decrease the novelty and the originality of the paper;

in the draft that I received, the journal editing instructions are not fully respected, especially at the references part (too big spaces between lines, not justified, etc. ). Please check and correct!

I suggest MINOR REVISION!

Author Response

Point 1: In the Abstract, but also in the Conclusion, replace "three series" with 32 quinoxaline-derivatives, since you mention 9 series in both parts;

Response 1: Thanks for your suggestion.

We have modified it according to your suggestion in the manuscript.

Point 2: Reformulate the phrase of lines 60-62

Response 2: Thanks for your suggestions. According to your suggestion, we have corrected it in lines 60-63.

Point 3: lines 68-70: What's the link to all mentioned before? Delete this part or explain it while linking it to the above mentioned;

Response 3: Thanks for your valuable advice.

In accordance with your suggestion, we have deleted this part, and the relevant references are added to lines 79-80 in the manuscript.

Point 4: line 76: replace "assume" with "assumed";

Response 4: We are very sorry for the mistake. We have corrected it in line 74 in the manuscript.

Point 5: Scheme 1: insert the structures of compounds 8 and 9;

 Response 5: Thank you for your good advice. As your suggestion, the structures of compounds 8 and 9 are inserted in Scheme 1.  

Point 6: line 127: replace "diemthyl" with "dimethyl";

Response 6: We are very sorry for the mistake. We have corrected it in line 125 in the manuscript.

Point 7: line 145: delete "strong";

Response 7: We are very sorry for the mistake. We have corrected it in line 143 in the manuscript.

Point 8: lines 148 and 150: replace "inhibition activity" with "inhibitory activity"

Response 8: We are very sorry for the mistake. We have corrected them in lines 146 and 148 in the manuscript.

Point 9: line 158: replace "capability" with "capacity";

Response 9: We are very sorry for the mistake. We have corrected it in line 155 in the manuscript.

Point 10: remake all the References part, since most of the articles cited are more than 10 years old and decrease the novelty and the originality of the paper;

Response 10: Thank you for your good advice. According to your suggestion, we have updated some old papers with recent ones in the manuscript.

Point 11: in the draft that I received, the journal editing instructions are not fully respected, especially at the references part (too big spaces between lines, not justified, etc. ). Please check and correct!

Response 11: We are very sorry for the mistake. According to your suggestions, we have modified the format, space, line spacing and other issues in the references.